# Optical Waveguides and Integrated Optical Devices for Medical Diagnosis, Health Monitoring and Light Therapies

**DOI:** 10.3390/s20143981

**Published:** 2020-07-17

**Authors:** Jiayu Wang, Jianfei Dong

**Affiliations:** 1Suzhou Institute of Biomedical Engineering and Technology, Chinese Academy of Sciences, Suzhou 215163, China; wjy174@mail.ustc.edu.cn; 2Department of Electronic Engineering and Information Science, University of Science and Technology of China, Hefei 230026, China

**Keywords:** waveguides, optical devices, photomedicine, sensing, implantable, wearable

## Abstract

Optical waveguides and integrated optical devices are promising solutions for many applications, such as medical diagnosis, health monitoring and light therapies. Despite the many existing reviews focusing on the materials that these devices are made from, a systematic review that relates these devices to the various materials, fabrication processes, sensing methods and medical applications is still seldom seen. This work is intended to link these multidisciplinary fields, and to provide a comprehensive review of the recent advances of these devices. Firstly, the optical and mechanical properties of optical waveguides based on glass, polymers and heterogeneous materials and fabricated via various processes are thoroughly discussed, together with their applications for medical purposes. Then, the fabrication processes and medical implementations of integrated passive and active optical devices with sensing modules are introduced, which can be used in many medical fields such as drug delivery and cardiovascular healthcare. Thirdly, wearable optical sensing devices based on light sensing methods such as colorimetry, fluorescence and luminescence are discussed. Additionally, the wearable optical devices for light therapies are introduced. The review concludes with a comprehensive summary of these optical devices, in terms of their forms, materials, light sources and applications.

## 1. Introduction

With the rapid rise of biomedical photonics since the 1980s, revolutionary photomedicine has emerged. As a new way for medical diagnosis and treatment, photomedicine combines optics, medicine and biotechnology together to study the structures and functions of biological tissues, and can further lead to physiological molecule detection. Thus disease diagnosis, treatment and theranostics in macro and micro scales can be realized [1]. For instance, photodynamic therapy (PDT) for curing cancers [2,3], photothermal therapy (PTT) for tissue ablations [4], photobiomodulation (PBM) for cell proliferation and antimicrobial activities [5,6] have already been utilized in clinical medicine. Significantly, non-invasive physiological signal detections including but not limited to blood glucose, blood oxygen and analysis of chemicals in the body fluid have been successfully achieved [7,8]. Besides, high-resolution optical imaging [9,10] and real-time sensing [11] are becoming more and more popular in human medical use based on sensing methods. The newly emerging neural regulation and modulation technique, optogenetics, also has a burgeoning demand for optical stimulators possessing simultaneous sensing functions, such as optical, electrical and microfluidic functions [12].

Biocompatible optical devices should provide non-invasive, minimally invasive and implantable treatment and diagnosis for the human body, while minimizing tissue reactions that could be harmful for the patient or impair the functionality of the device. To fulfill the requirement of biocompatibility of these optical sensors or devices, biomaterials composed of artificial macromolecules can be applied in photomedicine. The essential prerequisite of biomaterials is biocompatibility, which enables materials to perform with an appropriate host response in medical applications [13]. Generally, these biomaterials can interface with biological systems to evaluate, treat, augment or replace any tissue, organ or function of the body.

The biocompatible optical devices function as the media between light and tissues during the process of real-time sensing and therapy. In the past decades, the development of biocompatible materials and microfabrication techniques has endowed the biocompatible optical devices with more abundant and diverse functions [14,15,16].

In this review, biocompatible optical devices have been categorized into single waveguides, integrated implantable optical devices and integrated wearable optical devices. This review work is intended to link the multidisciplinary fields including materials science and manufacture processing, mechanics, optics, electronics, medicine and sensing methods. Typically, optical waveguides are fabricated based on biocompatible materials, such as glass, natural polymers, synthetic polymers and hybrid materials. These waveguides usually exhibit great mechanical properties and can propagate light efficiently. As for optical waveguides, diverse disciplines of the devices including the optical and mechanical properties, fabrication methods and optical structures will be emphasized. In the aid of optical sensing methods, integrated implantable optical devices can become chronic and stable tissue—device interface to support in-vivo medicine. Introductions to the implantable optical devices will feature the constructions of the optical sensing modules and their optical performance. Wearable optical devices can not only serve as platform for continuous and non-invasive molecular analysis but also function as core equipment of light therapies. The detailed descriptions of the optical sensing modules of the wearable devices will be discussed. Next, the clinical photomedicine including optical imaging, physiological signal detection, PDT, PTT and PBM will be introduced. The newly emerging optogenetics for manipulating cells and neural systems is also covered in this work.

This review is structured as follows: As the functional core, optical waveguides based on light sensing function will be discussed in the “Biocompatible optical waveguides” section. In the “Integrated implantable optical devices” section, light active and passive devices combined with optical sensing methods will be reviewed. Integrating wearable optical devices according to the optical detection methods will be introduced in the “Integrated wearable optical devices” section. In the “Photomedicine based on biocompatible optical devices” section, clinical applications including optical imaging, physiological signal detection, health monitoring, PDT, PTT and PBM will be reviewed, together with the optogenetics that can modulate neural activities. Conclusions will be drawn in the last section.

## 2. Biocompatible Optical Waveguides

Optical waveguides with ultralow transmission loss have been widely used in many fields such as biomedical applications, telecom and sensing. As in telecom, single-mode fibers (SMFs) made of silica glass play the core role in communication systems, promoting the connection of the world with highly-efficient network. In addition, the compositions of these fibers are a core and cladding. Nevertheless, these silica-based fibers are too hard to adapt with human tissues and may provoke harmful reactions and impairment of the fibers when used. Biocompatible optical waveguides can perfectly compensate for these defects, because they cannot arouse foreign body reactions, and are flexible and able to be biodegraded. The optical devices based on these biocompatible waveguides can be inserted into the human body with the great property of light transmission.

Biocompatible optical waveguides with light sensing functions are widely used in photomedical applications, such as PTT, PDT, optical imaging, physiological signal detection, and optogenetics. The waveguides made of glass-based materials, polymer-based materials and heterogeneous materials will be discussed in this section. The optical properties of these waveguides including optical loss and refractive index (RI) are summarized in detail, together with their fabrication processes, mechanical properties and applications.

### 2.1. Glass-Based Waveguides

Glass-based waveguides have become promising candidates for photomedicine, owing to the outstanding properties, such as the excellent transparency, high threshold to optical damage, low propagation loss and low cost [17]. Silica is the representative and dominating glass-based material to be implanted into biological tissues, photomedical applications which has several distinguished properties. Firstly, silica has a wide wavelength range, from ultraviolet (UV) to near-infrared (NIR). Secondly, silica has good mechanical performance, which is not easily broken or bent, and has chemical inertia that cannot undergo hydrolysis or enzymatic hydrolysis. Thirdly, many materials such as aluminosilicate [18], germane silicate and calcium-phosphate [14,19] can be doped into silica.

Through doping metal salt impurities into silica, the biocompatibility and biodegradability of synthetic materials can be achieved [17]. For example, a type of step-index fiber made of highly transparent calcium-phosphate glass can be used in optogenetics and optical sensing [20]. These fibers can exhibit optical transparency in a wide wavelength range, from UV to NIR, and have low intrinsic attenuation loss. During the manufacture of fiber drawing, the materials have shown thermal stability. The waveguides are also proved to be resorbable in biomedical environment; but the studies in biosensors still need deep excavation [20]. In addition to the fiber structure of waveguides, silica-based waveguides have been designed into arrays for applications in optogenetics, highly selective PDT and deep tissue imaging for diagnostics. As an example, Figure 1 shows a microfabricated array with microneedles characterized to provide several modes of optical excitation with deep penetration [21]. Notably, the width, length and tip taper angle of the array can be adjusted according to specific applications (e.g., optogenetics and PDT) [21]. Besides, combinations of light delivery and light sensing-based diagnosis have been demonstrated to be capable of performing theranostics [21].

However, most light guiding systems based on glass materials are not compatible with biological systems. Silica-based fibers may cause infection and immune reactions at an implanted site, resulting in inflammation and discomfort to patients [13]. To overcome this limitation, there exists continuing efforts to develop optical waveguides made of polymer materials.

### 2.2. Natural Polymer-Based Waveguides

#### 2.2.1. Silk-Based Waveguides

As a kind of natural material, the silk derived by silkworms or spiders has been widely exploited to form fibers because of its excellent mechanical properties, great biocompatibility and tunable time of biodegradation. Good transparency of this material makes itself an effective channel for light sensing. Moreover, soft and flexible silk-based fibers have the same mechanical properties as human tissues, and can hence reduce the damage during implantation [22,23,24].

Recently, a class of fibers made of silk fibroin solution has been designed through direct inking [25]. The silk waveguides have been demonstrated to be capable of guiding light in biological tissues with low optical loss, which can develop new avenues of applications in optical imaging and phototherapy. The silk film surrounded by a silk hydrogel cladding has a smooth surface and high transparence, as shown in Figure 2a. Notably, more complex patterns of these types of waveguides can be obtained by the manufacture process of direct inking [25]. The femtosecond direct laser writing (fs-DLW) to fabricate waveguides made of silk fibroin has been reported [26]. Through this way, biocompatible waveguides of 3D structures with minimal thermal damage can provide a versatile platform for photomedical applications ranging from implantable optical devices to biosensors [26]. By means of genetic engineering, recently, a new class of unclad fibers made of recombinant spider silk (see Figure 2b) has been fabricated [27]. Compared with optical waveguides made of regenerative silkworm protein, the recombinant spider silk-based waveguides have higher RI and a smoother surface to guide light into deep tissues with low loss. Incorporating excellent optical, biocompatible and biodegradable properties, this optical waveguide is a prospective strategy for photomedical applications such as PBM and PDT [27].

Due to the different stiffness of human tissues, the silk hydrogel with tunable mechanical properties can provide a feasible choice for different photomedical applications. In one study, the stiffness of the in-vivo spun fibers has been improved by inserting spider silk sequences into Bombyx mori silk [24]. In addition, silk-elastin-like proteins (SELPs) have been extensively studied to be leveraged for photomedical applications. What distinguishes SELPs is the convenience to design intended forms and to function by varying the silk-to-elastin ratios. By adjusting the ratios, materials with specific temperature-based swelling and mechanical properties can be achieved [28].

#### 2.2.2. Cellulose-Based Waveguides

Cellulose is a linear condensation polymer that has been widely utilized in making optical fibers due to the outstanding optical and biological properties. Cellulose-based fibers have a wide spectral transmission of light and high transparence, along with unique permeability to water and ions. Interestingly, fibers with higher cellulose content can exhibit higher tensile strength and modulus [31]. Such biocompatible and biodegradable fibers are usually fabricated into microstructured fibers, possessing the capacity to integrate multiple functions used in biosensing or phototherapy.

For example, a biodegradable optical fiber made of cellulose that embodies optical, microfluidic and drug release functions has been fabricated via thermal drawing [29]. The geometry presents a porous double-cladding, in which the inner core is suspended in the middle of an outer cladding by the intact powder particles. The clear internal structure profile can be clearly seen in Figure 2c. Sensing by using these fibers relies on changes of fiber transmission properties when fiber pores are filled with analyte. The easily tailored design of the structure of these fibers can support many applications such as optical imaging, PBM and biosensing. An intriguing possibility is to incorporate optical, sensing and medical treatment functionalities into the same biocompatible fiber to create a highly integrated and self-sufficient medical system for theranostics [29].

Another incorporated fabrication process with co-rolling of plastic films, powder-filling and solution-casting has been investigated to manufacture fibers with functionalized hollow microstructures [30]. Particularly, microstructured fibers with functionalized hollow microstructure can be used in vivo for real-time detection of biomolecule binding events. Hollow core fibers with multilayer can be used in vivo for delivering medically important high-power laser light. Step-index fibers with multicore can be used in vivo for fluorescence microscopy by delivering high power light in a smaller core and collecting fluorescence in a larger core with higher numeric aperture (NA). Finally, the in-vivo direct use of biopolymer optical fibers promises a more compact alternative to endoscopes [30]. Figure 2d shows the transmitting light profiles of the diverse cross-sections of cellulose-based fibers. Made of biomaterials with great physical and optical quality (e.g., poly(l-lactic acid) (PLLA), polycaprolactone (PCL) and cellulose derivatives), these fibers can be used in in-vivo optical imaging and light sensing [30].

### 2.3. Synthetic Polymer-Based Waveguides

#### 2.3.1. Polylactic Acid (PLA)-Based Waveguides

As a type of liner aliphatic polyesters, polylactic acid (PLA) is the most abundant polymer among all the synthetic polymers. PLA can be divided into poly (d-lactic acid) (PDLA), poly (l-lactic acid) PLLA and poly (d,l- lactic acid) (PDLLA) according to the composites of different mixture of D and L isomers, which are optically active stereoisomers called dextro- (d-) and levo- (l-). Macroscopic mechanical properties ranging from softness to stiffness can be obtained through controlling the molecular weight of the polymer. PLA has great biocompatibility, biodegradation and high transparency, which enable its applications in photomedicine and light sensing [32].

In one study, an unclad PDLLA-based optical fiber manufactured by granulate melting and heat drawing has been demonstrated to take two to three months to degrade in the human body [33]. Particularly, these fibers have an attenuation coefficient of 0.11 dB/cm at 772 nm, which is the lowest loss reported so far for optical polymer fibers. In addition, cases regarding the biocompatibility of PDLLA in vivo have been reported, where no clinical signs of foreign body reactions have been discovered. This type of fibers can be used to deliver light in vivo for light sensing applications [33]. Another type of comb-shaped slab waveguide (see Figure 3a) is fabricated by using melt pressing, solvent casting, laser cutting and ultraviolet-induced crosslinking techniques, which can be used for long-term light delivery [34]. This type of waveguides can allow the watertight crosslinking between the tissues to be formed in situ; the waveguide need not to be removed from the site after the processing, as it is eventually biodegraded and absorbed by the tissue. Implantable waveguides can induce deep tissue stimulation by low-level light whose irradiance ranges from 5 $\mathrm{mW}/{\mathrm{cm}}^{2}$ to 5 $W/{\mathrm{cm}}^{2}$ [6] and offer new strategies for light sensing-based diagnostics, therapeutics and theranostics [33,34].

#### 2.3.2. Polyethylene Glycol (PEG)-Based Waveguides

PEG is a kind of polyether, which exhibits great optical and mechanical properties, non-toxicity and water-solubility, along with biocompatibility and biodegradation. PEG-based optical fibers have been widely applied in photomedicine such as diagnostics, therapies and theranostics [38].

In one study, PEG hydrogels are used to fabricate slab-shaped film encapsulated with cells for optical sensing and light therapy [39]. During the fabrication, UV-induced polymerization happening upon PEG-diacrylate (PEGDA) solution can be achieved in the custom-made glass mold [39]. Notably, before photo-crosslinking, the coupled light source must be concatenated to the PEGDA preform. This flexible fiber can be implanted in vivo and serve as an optical communication channel between the encapsulated cells and an external light source. The in-vivo implantation experiment has shown its great ability in real-time green fluorescent protein (GFP) sensing and optogenetics [39]. Another step-index fibers can propagate light into deep tissues, which has been manufactured to ensure highly efficient light transmission into deep subcutaneous tissues via photocrosslinking and dip-coating [35]. As shown in Figure 3b, when blue laser is coupled, the dye-doped region emits yellow fluorescence from excited rhodamine-6G molecules. By incorporating various functional fluorophores and nanoparticles into their porous structure, the feasibility of in-vivo optical sensing and light-induced therapy can be realized [35].

Incorporated with various biomaterials, optical sensing system with high sensitivity based on hydrogel fibers has shown the potential for continuous in-vivo glucose monitoring [40]. For the composites of this step-index fiber, the core is manufactured of a poly(acrylamide-*co*-poly (ethylene glycol) diacrylate) p(AM-*co*-PEGDA); and the cladding is manufactured of Ca alginate. Importantly, 3-(acrylamido)phenylboronic acid (3-APBA) molecules are covalently incorporated into the core, which can combine with the diffused glucose molecules to change the hydrogel density. The variation of hydrogel density can change the RI that affects light propagation through the hydrogel fibers. Hence, the change of RI of core can be utilized to quantify glucose concentrations. The fabrication process contains several important steps including UV-light exposure and dip-coating. Additionally, these highly flexible fibers with low optical loss can be functionalized with chelating agents, proteins and channel-based membranes, as to be responsive to diverse analytes for optical sensing [40].

Moreover, an approach involving PEG hydrogel incorporated with cellulose particles as a reinforcing composite has been designed for in-situ photopolymerization and monitoring in a minimally invasive manner [41]. For the photopolymerized process, the detected size of the samples is correlated with the fluorescent signal collected by the probe tip inside a bovine caudal intervertebral disc. In addition, the probe tip can be presumably positioned in vitro based on the collected fluorescence and Raman signals. The UV-induced manufacture procedure through the optical fiber inside tissues has shown clinical possibilities such as health monitoring [41].

#### 2.3.3. Polydimethylsiloxane (PDMS)-Based Waveguides

Polydimethylsiloxane is a kind of silicon-based organic polymers, and is particularly known for its flexible mechanical property to be biocompatible with tissues. PDMS is optically transparent with high RI, bioinert, non-toxic and biodegradable [36,42], which can be applied in phototherapy and optical sensing. Figure 3a shows a sample cylindrical profile PDMS waveguide.

For example, the PDMS-based double-layer waveguide used to transmit light through skin tissues for powering implantable electronic array has been made via curing and dip-coating [43]. Notably, this type of waveguides has efficient effect of collecting and delivering light for powering electronics, which can generate much smaller tethering forces compared to metal wire interconnects [43]. Another example demonstrates a stretchable PDMS-based fiber for deformation sensing because of the excellent optomechanical property and reliable performance during repeated exposure to the strains [44]. The coextrusion process is devised to form core/cladding fibers maintaining their mechanical integrity for strains up to (545 ± 35)%. This new type of fiber can be utilized as health monitoring sensor with great mechanical flexibility [44]. Notably, linearly tapered optical waveguides made of PDMS have been fabricated via mold casting [45]. These side-emitting waveguides can be warped around the scleral for uniform light delivery. These waveguides are well suited for accessing the entire equatorial circumference owing to their long light-delivery length (>7 cm), small thickness (<1.5 mm), elasticity and flexibility [45].

#### 2.3.4. Polyacrylamide (PAM)-Based Waveguides

PAM is a kind of linear polymers with high hydrophilicity. PAM-based optical waveguides have been widely used in medical devices for phototherapy or optical sensing. The applications rely on their outstanding properties including but not limited to bio-inertness, biocompatibility, long span in-vitro, optical clarity and mechanical flexibility.

In one study, the optical sensor based on PAM waveguide, aiming at monitoring the wound healing process, has been developed [46]. Briefly, the fabrication process of the hydrogel waveguide takes two steps. The first step is to deposit the responsive hydrogels that are incomplete cross-linking polymer mixture onto the substrate. The second step is to silanize the substrate before UV-light induced photopolymerization. Sensing is based on changes of RI of hydrogel which responds to variation in pondus hydrogenii (pH) and C-reactive protein concentration. The waveguide can be integrated into wound dressings, forming an optical sensing device that is connected to optical guidance and electrical powering modules. With the existence of these combined chemical and optical sensing measures, a new perspective about optical devices for health monitoring has been opened up [46].

Besides the application in health monitoring, an alginate-PAM hydrogel optical fiber, which is low-modulus and highly stretchable, has been investigated for in-vivo optogenetics modulation [37]. Through the polymerization and cross-linking of alginate-PAM precursor in different tube molds, the optical fibers with highly efficient transmission can be fabricated. Figure 3d shows the blue light-emitting diode (LED)-coupled fiber with uniform irradiation [37]. Additionally, a new class of PAM microfiber doped with Au nanorods used for relative humidity sensing is presented [47]. Through direct drawing from PAM aqueous solution followed by Au rods deposition, researchers have obtained a moisture sensitive microfiber capable of forming whispering gallery modes (WGMs) and localized surface plasmon resonance (LSPR). With relative humidity (RH) increasing, more water molecules can diffuse into the polymer microfiber to induce surface plasmon resonances, which can change the intensity of output light. Through this way, the scattered spectra that reveals the change of RH of the microfibers can be shown. By analyzing these changes, the diagnosis of respiratory diseases can be achieved [47].

### 2.4. Heterogeneous Materials Based Waveguides

More and more heterogeneously integrated optical waveguides have been widely developed in therapies, optogenetics and theranostics. What distinguish them are outstanding mechanical flexibility to adapt to deep tissues, biocompatibility and great functionalization of their materials. As novel waveguides, benefits can be brought to photomedicine such as PDT, optical sensing and optogenetics [16].

Such multifunctional optical fibers can accomplish efficient light guidance and possess superior optoelectronic sensing attributes. A novel hybrid fiber allowing for fluorescent imaging has been demonstrated via thermal drawing process and solution-based nanowire growth approach [48]. Integrated with a single crystal semiconductor nanowire-based module at the tip, these advanced optical fibers exhibit great optoelectrical properties. These fibers containing the SnZn electrodes, polycarbonate (PC) films cladded by phenylsulfone (PSU) films and the polycarbonate (PC)/polymethyl methacrylate (PMMA) solid rod will provide a tool for health care, fluorescent imaging and biosensing [48].

Due to the development of optogenetics in a couple of past decades, the complexity of neural activity research has led to the ascension of a new type of fibers that can be multifunctional (e.g., simultaneous optical, fluidic, electrical and chemical functions) [49,50]. For example, researchers have made a class of preform consisting of mixed materials including poly(etherimide) (PEI), poly(phenylsulfone) (PPSU), polycarbonate (PC) and cyclic olefin copolymer (COC), a polymer composite (conductive polyethylene (CPE)) and a low-melting-temperature metal (tin (Sn)) [51]. Significantly, all these materials have the same glass transition and melting temperatures, allowing for forming uniform fibers, when heated and stretched via thermal drawing. Two designs of neuron probes made of COC, PC and CPE, which have different optical losses, are shown in Figure 4a, together with their light guiding cross-sections. Further, multielectrode fiber probes consisting of a PEI core doped with Sn and PPSU cladding have been fabricated, showing a great performance in electrical signal recording [51,52].

Moreover, a highly stretchable step-index fiber made of SEBS (a linear triblock copolymer based on styrene and ethylene/butylene groups), with an optimized ratio between soft poly (ethylene-*co*-butylene) and hard phase (polystyrene, *n* = 1.52) core and Geniomer (a block copolymer made up of soft and hard segments (siloxane/aliphatic isocyanate, *n* = 1.42)) cladding, has been developed [53]. Another prototype of fibers with more a complex structure of PC core and SEBS cladding is also manufactured through thermal drawing, which can provide a tool for phototherapy, optogenetics and other health-care applications [53].

To achieve electrophysiological recording and optical neuromodulation co-instantaneously, a class of hybrid fiber maintaining low optical loss has been characterized as a flexible polymer-based fiber coated with silver nanowires meshes [54]. In this design, silver nanowires (AgNWs) and PDMS are respectively dip-coated as the cladding in sequence for electrophysiological recording and protection. The structure of this flexible fiber and the light transmission therein is shown in Figure 4b. These fibers may, in the future, be tailored to address fundamental problems in spinal cord or visceral organ neurophysiology [54]. Typical fabrication process involving the assembly of the preform and fiber drawing process is illustrated in Figure 4c [54]. Besides, quantum dots (QDs)-doped polymer microfibers have growing wide applications in photomedicine due to the excellent properties, such as adjustable emission wavelengths and photochemical stability. For instance, a novel QDs-doped polymer microfiber (Figure 4d) has been fabricated via thermal drawing using the polystyrene mixed with the PMMA [55]. Under light excitation from the focused laser, the polymer fiber can function as optical sensors, and can serve as a waveguide for phototherapy [55].

### 2.5. Key Principles of Waveguides

In order to implement better functionalities in photomedicine, the most important optical properties of the optical waveguides are optical loss and RI. Optical loss is the attenuation of the light power through an optical material per unit length, which directly affects the transmission distance. RI is the ratio of speed of light in vacuum to the phase velocity of light in a material or a medium. Whether using the side-emitting fiber or the straight-emitting fiber, to achieve total internal reflection, the RI of the inner layer of the waveguide must be higher than that of the surrounding materials. Novel structures of the core accompanied by cladding have been widely investigated to fulfill light transmission into deep tissues. Moreover, the transmission capability of optical fiber is greatly affected by surface smoothness and processing uniformity of the fiber. Mechanical properties of waveguides determined by the materials and fabrication processes also play an important role in photomedical applications. Finally, the key principles of the waveguides reviewed in the section are summarized in Table 1 and Table 2, with, respectively, the optical and mechanical properties.

## 3. Integrated Implantable Optical Devices in Photomedicine

Implantable devices combined with optical sensing modules can be used in many medical fields, such as drug delivery, cardiovascular healthcare and biomarkers sensing. These devices can provide high pinpoint spatial accuracy, and support efficient light delivery due to the integration of multifunctional modules such as optical, electronical, mechanical and chemical modules. Therefore, the integration of multiple functional features becomes a key requirement and a major challenge in device fabrication. In recent years, the development tendency of implantable optical devices has been directed to novel devices with complex constructions based on diverse materials [56,57,58,59]. Advanced material designs and integration strategies can provide diagnostic or therapeutic functionality for biocompatible, even biodegradable optical devices. These devices can be divided into two types of passive and active optical devices, depending on whether the light source is internal and spontaneous. As the media between light and tissues, optical sensing modules in these devices function as indispensable components. Applications of these optical devices have found enormous potentials in accomplishing diagnosis, treatments and theranostics. The emphasis reviewed here will lie in the constructions of the optical sensing modules, along with their implementations. At the end of this section, the optical modules and fabrication process of integrated implantable optical devices have been summarized in Table 3.

### 3.1. Passive Optical Devices

Without inner light sources, passivity-based optical devices must accept the outer excitation light to trigger the photomedical process, which can be controllable and accurate outside the biological bodies. Applications of these devices can be seen in many places such as optogenetics, light therapies and optical sensing.

Optogenetic stimulators can support optogenetics based on various opsins or light-sensitive proteins, which connect neural science and optical sensing. Optogenetic stimulators activated by an external optical stimulus have been widely investigated. For example, a kind of multifunctional neuron probes containing microfluidic channels and eight electrodes of every four shanks, along with a SU-8 (a negative photoresist)/glass optical waveguide on shank 1 (S1) has been fabricated [60]. The optical fiber can be placed in a U-groove pattern on the body of the probe array, and fixed with UV-curable epoxy. Figure 5a shows the transmission of blue light and the dual characteristics of optical and microfluidic structures in details. As a result, sufficient light intensity for optogenetics stimulation can be achieved at the end of the waveguide. In addition, the micro structure of shank can efficiently reduce tissue damage when inserted into the brain. The multi-shank multifunctional probe can provide simultaneous recording of neural signals in different regions, modulation of long-range neural circuits through optical stimulation, and drug infusion in the deep brain region of small animals [60].

In the process of intravascular theranostics, a biodegradable system incorporated with multifunctional electronic and therapeutic utilities has played a significant role with the assistance of outer excitation light sources. As an example, a type of bioresorbable electronic stent (Figure 5b) containing nanomembrane-based flexible flow/temperature sensors and memory storage devices, anti-inflammatory nanoparticles and drug-loaded core/shell nanospheres can be activated by an external optical stimulus [61]. As shown in Figure 5c, this endovascular implant incorporates a bioresorbable flow sensor for blood flow monitoring and a temperature sensor for blood sensing. Besides, a microprocessor module with wireless communication function will accomplish the analysis. Therapeutic nanoparticles (ceria nanoparticles and gold nanorod core/mesoporous silica nanoparticle shell (AuNR@MSN)) loaded with drugs are bound in the PLA layers of the stent coated with dissolution-rate controllable oxide/polymer. Under NIR light radiation, the heat generated from gold nanorods (AuNRs) can be transferred to drug-loaded mesoporous silica shell which facilitates desorption and diffusion of the loaded drug. Importantly, the NIR-light emitted from the optical fiber can be guided to the endovascular location for dose-controlled drug delivery. In addition, through minimally invasive or noninvasive optical control, the AuNRs can be heated to cause cell injury for PTT. The entire system has been experimented keeping efficient functions in-vivo during a week, which has provided new opportunities for the integration of sensors, data storage elements and optically responsive therapeutic nanoparticles for bioresorbable endovascular implants [61].

On the other hand, incorporated with the characteristics of painless insertion for drug delivery, tunable needle height and efficient light sensing, microneedles coupled with light sources are promising in the field of PTT. Recently, a new class of dissolving graphene/polyacrylic acid (PAA) transdermal microneedles that can simultaneously deliver chemical drug and transmit NIR light has been manufactured [62]. PDMS mold can be utilized to manufacture these microneedles via a two-step casting process, including mold injection and centrifugation. Due to the graphene dispersant in the PAA matrix, the microneedles can achieve improved mechanical strength and a great potential for PTT, which is evaluated under the NIR light irradiation for 2 min from 1 cm above a 24-well plate. After NIR irradiation, the graphene incorporated in the microneedles can convert light energy into thermal energy. Due to easy dissolution of the PAA, the microneedles can quickly deliver drugs to the skin. Significantly, the rapid delivery and biological safety of the water-soluble PAA matrix indicate the benefits of the PTT application in vivo. Control experiments have been conducted to research the dissolution of microneedles, as shown in Figure 5c [62]. Moreover, a PLA-based optical microneedle array has been demonstrated for percutaneous light delivery in PDT [63]. This array consists of a microneedle array consisting of 11-by-11 needles and a micro-lens array focusing incident light into each needle. Further, the PLA microneedle array is fabricated via press melting in a PDMS mold and the micro-lens array made of fused silica. Figure 5d shows the transmission pattern as emitted from the top end; and the transmitted light is relatively uniform. This novel device can deliver light into 2.5 mm deep tissue, leading to an eight-fold enhancement. This optical device with great function of light transmission can be utilized to improve the efficacy of phototherapies, such as blue light therapy for antimicrobial treatment and photodynamic therapy for cancerous lesions [63].

### 3.2. Active Optical Devices

For overcoming the drawbacks of connecting with the external light sources of passive optical devices, gradually near-field wireless optoelectronics have emerged. Generally, most of these kinds of devices are applied in the field of optogenetics. By means of transiting light into specific neurons with low loss and high precision, at the same time, potential sensing can be carried out.

Recently, a complicated device (Figure 6a) with multi-functional layers has been fabricated [64]. Built on a substrate of polyimide (75 mm thickness) in an overall planar geometry, the injectable needle with bilayer encapsulation enables robust operation. Conventional fabrication techniques including electro-deposition, photolithographic technique, etching, laser cutting and dip-coating are used during the process. The micro inorganic light-emitting diode (m-ILED) on the tip of the needle can create different wavelengths including UV, blue, green-yellow and red light to activate opsins and light-sensitive proteins. When m-ILED delivers the light stimulation, the indictor LED will be lightened. For in-vivo applications, wireless photostimulation of dopaminergic neurons in the ventral tegmental area has been demonstrated through characterization of Channelrhodopsin-2 (ChR2) expression in the nucleus accumbens (NAc) or ventral tegmental area (VTA). This device possesses a wide applicability across the optogenetics research community [64]. In another study, a novel m-ILED array integrated with microfluidic channels and an optical waveguide is manufactured on a PDMS substrate [65]. As shown in Figure 6b, the m-ILED is installed on the filament of polyethylene terephthalate to ensure spatiotemporal control of neural circuit with minimal insertion. The fabrication process begins with the mold curing of the channel patterns in a PDMS layer followed by pressing with a glass slide. Then another PDMS-coated polycarbonate (PC) membrane is oxygen plasma-treated, together with the former layer. Specifically, optical manipulation of projections from the VTA into the NAc can elicit place preference behaviors that can be blocked, in a temporally precise programmable manner, by site-specific infusion of a dopamine receptor antagonist. These optofluidic neural probes can deliver multiple types of pharmacological agents and monochromatic light to discretely targeted regions of the deep brain [65].

Furthermore, the highly integrated optogenetics system (Figure 6c) allowing for a battery-free, fully-implantable optofluidic sensing cuff installed with μ-inorganic light-emitting diode (μ-ILED) has been developed [66]. In total, six well-organized subsystems are assembled into a system. For the fabrications of the soft cylindrical nerve cuff, firstly, a PDMS preform is formed from PDMS precursor; the next step is thermal drawing, and a strip of PDMS can be formed via corona treatment; finally, the shaped cuff is achieved through releasing the pre-strain. Through using the magnetic inductive coupling matched to near-field communication (NFC), enough energy can be obtained to trigger drug delivery and to lighten the μ-ILED. Coupled with blue light, the system has been illustrated to be capable of activating the neurons in nociceptors of the ChR2 expressed mice to induce significant aversive behavior. Another experiment has demonstrated temporal drug delivery to the sciatic nerve in vivo. The full optofluidic system, showing chronical stability and non-toxicity during a two-week implantation, can improve the tool set available for advanced manipulation of peripheral nerves in awake, freely moving animals [66].

## 4. Integrated Wearable Optical Devices

Optical devices for diagnosis based on sensing methods can provide instantaneous and continuous detection of relevant biomarkers in the human body, as to diagnose diseases and monitor personal health conditions. Highly efficient integrated optical devices can combine light delivery/collection with intrinsic sensing functionality. Based on internal reaction principles, these devices can be classified into colorimetric, fluorescent and luminescent optical devices, as to retrieve analyte information related to human physiological situations [7,8,16,67,68,69,70,71,72,73,74]. The advances in optoelectrical materials, wireless communication and micromachining technology have promoted the trends of wearable devices towards real-time, continuous monitoring of health status, as well as light therapies for wound healing, tumor treatment and antimicrobial treatment. Importantly, the working principles and constructions of optical sensing modules will be presented in this section. In addition, the operation situations of the optical devices will be briefly discussed, along with the fabrication processes and applications of the different optical devices for diagnosis and therapy. At the end of Section 4.1, the optical sensing modules and applications of integrated wearable optical devices for diagnosis have been summarized in Table 4. The optical sensing modules and applications of integrated wearable optical devices for therapy have been summarized in Table 5 at the end of Section 4.2.

### 4.1. Integrated Wearable Optical Devices for Diagnosis

#### 4.1.1. Colorimetry-Based Optical Devices

Colorimetry-based optical devices combine soft, flexible properties and thin layouts, allowing robust and nonirritating interfaces with the epidermis. Multifunctional components including colorimetric layer, microfluidic construct and optical analysis module compose these wearable devices. As the most widely used optical sensing methods in wearable optical devices, colorimetry refers to presentation of the changed color of sensing elements reacted with analyte. The quantified concentration of target analyte can be achieved through measuring the absorbance of sensing elements. Importantly, the modified substrates or colorimetric chemical reactions are pivotal factors of the optical specificity and sensitivity. In addition to wearable conformability and highly efficient operation, these devices using colorimetry method do not require pretreatment or calibration.

Colorimetry-based optical devices are adequate for many photomedical applications such as biofluid sensing for biomarkers including electrolytes, small molecules and proteins detection. For example, a type of colorimetric optical devices mounted on the wrist to measure sweat pH has been developed [75]. The integrated red−blue−green (RGB) reference markers in the assays can provide accurate color analysis that corresponds to sweat pH level. The color analysis displays a linear response across the physiologically relevant range of glucose concentrations [75]. Embodiments mounted directly on the skin can allow much more analysis of sweat. In one study, a soft, flexible and stretchable microfluidic system integrated the PDMS-enclosed microchannels has been designed [76]. PDMS manipulated utilizing soft lithography can define the microfluidic network which contains functionalized channels and reservoirs for sweat capture, routing and storage with spatially separated regions. Through photographing the color reaction by smart-phone, contents of key markers including glucose, lactate and pH can be detected accurately. Quantitative paper-based colorimetric assays embedded in the reservoirs afford assessment of pH and the concentration of lactate and glucose. During the enzymatic reactions, the induced color change of detection reservoir correlates with the concentration of selected essential biomarkers. Further, the ultrathin NFC electronics integrated on the top of the microfluidic system can enable wireless communication to external phone, thus direct electronic readout of quantitative information can be screened [76]. The colorimetry-based optical devices discussed here can represent versatile platforms for diagnosing health status.

In addition to application in sweat analysis, the optical detection strategy is also explored toward tear glucose monitoring. Recently, a new class of colloidal crystal array (CCA) embedded in a hydrogel matrix with attached contact lens has been investigated for glucose sensing [77]. Correlation of glucose levels in tears to those blood samples can enable a noninvasive means to manage diabetes with the aid of rigid gas-permeable (RGP) contact lens. Through the poly (vinyl alcohol) (PVA) gelation, polystyrene (PS)-based CCA can be designed on the irregular surface. During the analysis of physiological tear glucose concentration, the diffracted wavelength has a relative linear correlation to glucose concentration due to the dielectric periodicity of the PS particles. The diffraction wavelengths result from the Bragg stack nanostructure of the PS CCA and 4-boronobenzaldehyde (4-BBA) modified PVA, whose interval was called the photonic band gap, a periodic permutation that is an integer multiple of wavelengths of visible light. According to Bragg’s law, the changes of PS particle space can lead to the shift of the diffraction wavelength. These devices with excellent biocompatibility have shown promising potential of promoting the development of glucose detecting devices [77].

#### 4.1.2. Fluorescence-Based Optical Devices

A main drawback of colorimetry-based devices is that colorimetric assays exist only for a relatively narrow range of biomarkers. A fluorescence-based analysis of biomarkers can represent a complementary approach due to the short response time, minimal instrumentation requirements and low cost. This method is based on a change in the fluorescence intensity of an optically active molecular fluorescent dyes in response to a concentration of analytes variation. Importantly, the detection of these substances in tissue fluids is easier achieved through the fluorescence-based method. Fluorescent derivatives containing chemical reaction groups are usually required to react with the physiological analytes for molecule identification and detection. Many wearable optical devices utilize the fluorescent method to detect the concentration of biomarkers, as to evaluate the health status.

For example, a type of wearable multifluidic platform with the reservoirs to store the sweat (Figure 7b) from human skin has been designed [78]. The harvested sweat is spontaneously routed to serpentine channels connected with reservoirs that contain colorimetric assay reagents. The concentration of sweat analytes such as chloride, glucose, lactate, and pH can be measured. During the process of analysis, an optical module containing LED, filters and camera can capture fluorescence imaging. The LED coupled with the filter can emit light to the probes, meanwhile, the imager can detect the fluorescence. Different colors related to different concentrations of the biomarkers can be seen on the smartphone. Fluorescent signals from every microreservoir correlate quantitatively to the concentration of the target analyte, which can reveal accurate health status. Figure 7a illustrates the entire system of sweat sensor worn on the human skin [78].

Besides the analysis of sweat, based on the changes of fluorescence intensity, these devices can incorporate modules of light sources, polymer optical fibers and photodetectors to detect pH concentrations. In one study, a type of fluorescence-based optical devices implemented on a microfluidic chip was designed to measure pH concentrations [79]. The fluorescence dye named 8-hydroxypyrene-1,3,6-trisulfonic acid trisodium salt (HPTS) is biocompatible and photostable. This type of dye can be excited at 405 or 407 nm and has an emission wavelength at 520 nm. Optical fibers have been coupled with a 445 nm blue laser to transmit the excitation light to the photodiode (PD) detector. The fluorescence intensity is highly sensitive to pH that is correlated to resulting voltage at PD. Notably, a laser cutting method has been used in fabricating the microfluidic channels of PMMA-based chip [79].

#### 4.1.3. Luminescence-Based Optical Devices

Luminescence-based optical devices are especially attractive due to their high sensitivity and selectivity. The mechanism of luminescence-based optical devices is that the intensity variation between the emitted light and the received light is related to the concentration changes of analyte. With the aid of these devices, the analyte, such as oxygen, halides and various metals, can be measured with great performance.

In one study, a flexible oximeter array (Figure 7c) with silicon-integrated circuits has been developed to monitor a pulsatile arterial blood signal for diagnosis [80]. OLEDs are suitable for wearable display applications because they are flexible and ultrathin. The OLED arrays are fabricated on indium tin oxide (ITO) patterned polyethylene naphthalate (PEN) substrates. Importantly, four red OLEDs and four NIR OLEDs are chosen to emit light because the molar extinction coefficients of hemoglobin (Hb) and oxyhemoglobin (HbO_2_) vary appreciably over the visible and NIR spectrum. During the detection process, the red LED coupled with PDs can measure the changes of blood hemoglobin absorbance in the blood vessels by infrared emission module and receiving module; the green LED coupled with PDs can measure the density of blood in the blood vessels as it pulsates based on the changes in transmittance. Besides, other eight organic PDs arrays for collecting reflected signals are printed on top of planarized PEN substrates via blade-coating techniques [80].

In another study, a type of ultrathin skin-like systems has been used in pulse oxygenation measurement based on the principle of photoluminescence [81]. After light transmission in the body, the red and green light-emitting from the PLEDs can be detected by the OLED. Through this way, the voltage related to light intensity can be displayed. Figure 7d shows a photograph of a finger wearing the ultra-flexible organic optical sensor. Owing to the extremely flexible, compact and efficient optical structure, the device with high precision in monitoring the situation of the patients can assist the treatment of disease. Notably, laminated on the pre-stretched acrylic tape–silicone rubber sheet, the center OLED is neatly accompanied by two PLEDs [81].

### 4.2. Integrated Wearable Optical Devices for Therapy

#### 4.2.1. Passive Optical Devices

In the aid of coupled light sources such as LED, carbon arc lamp and lasers, passive optical devices can be used in many applications including cancer treatment and cell metabolism enhancement. Incorporated with biocompatible materials and micromachine technology, these passive optical devices with unique and specific structure can provoke uniform and standard-compliant light to anyplace across the human body with great performance and wearing comfort.

As a type of passive optical devices, textile fabrics with three-dimensional (3D) conformability can be worn on the human body, being ideal substrates to fulfill specific photometric requirements. Through making certain features in the core or at the core–cladding interface of the fibers, the polymer optical fibers can emit light sideway to form large-area 3D illumination. In one study, researchers have designed a luminous flexible fabric device integrated cotton yarns and PMMA-based optical fibers for phototherapy [82]. Notably, a V-grooved structure on the surface of the fibers can enable a part of light transmitting in the fiber to refract out, because these light rays cannot satisfy the requirements of light travelling in the fiber. These fibers with excellent optical properties have been put together and coupled with one LED by an optical adhesive, exhibiting an appropriate wavelength of 630 nm to activate collagen production. As shown in Figure 8a, these step-index fibers are woven as the weft yarn with cotton yarns, which can form a plain weave fabric structure. Such two layers of the same fabric, encapsulated by a black foam and connected with electrical circuit and LED, are utilized to guarantee uniform irradiation. Besides excellent optical properties, this flexible fabric device shows stable thermal properties, and can be a patch during light irradiation process [82].

Another type of flexible passive optical device is an optically transparent membrane with conformity to non-planar geometry. Typically, these sheet-like devices integrate multilayers of polymer waveguides where appropriate micro-optical features can be designed to collect or illuminate light. For example, a type of two-layer non-rigid sheet made of PDMS has been fabricated via soft-lithography techniques [83]. As shown in Figure 8b, the top layer with feature of micro-lens can collect ambient light over a relatively large area. The collected light can be transmitted to the bottom layer where light can exit the sheet. Notably, in order to form a core/cladding structure, the refractive indexes of the stacked layers made of PDMS are different. A micro-lens engraved on the top later can accurately focus the light rays onto the center of the micro wedge features embedded in the bottom face of the second layer. The appropriate angles of the micro features enable total internal reflection in the second layer, which can redirect the light rays out of the sheet in the terminal end. This type of devices can emit light in diffused region proportional to the intensity of light received by the concentrator region, which has great potential in phototherapy [83].

#### 4.2.2. Active Optical Devices

Active optical devices can directly offer specific and moderate light doses for photomedicine application without the outer coupled light sources. Interestingly, these light sources are both energy sources and sensing agents. Typically, wearable light patches can be used easily in any body location during free movement, which incorporate micron LEDs printed on flexible polymer substrate and can be adhered to the skin by hydrogels. Significantly, LED is currently the most efficient and energy-saving artificial light source. Other than employing discrete emitters imbedded in a substrate, there are a variety of technologies being developed that involve self-emissive devices [84,85]. For example, devices have been described that use LED arrays, polymer light-emitting diodes (PLEDs), thin film flexible electroluminescent sources (TFELs) and Q-dots LED [86]. Even organic light-emitting diodes (OLEDs) which extend the possibilities of phototherapy have been widely used, since OLEDs can emit light uniformly from a flexible surface [87,88].

Recently, a novel printed LED light patch for PBM has been investigated via roll-to-roll printed micro LED technology [89]. As can be seen in Figure 8c, the patch contains blue micro LEDs printed on micro polyester film. This film is laminated to a flexible circuit board. Ultra-flexible hydrogels laminated on the bottom of the patch can be adhered well to the human skin and enable efficient transmission. Notably, through using a layer of cadmium free quantum dot film or by printing LEDs of different wavelengths, the patch can provide multiple wavelengths of spectrum evenly over the illumination surface. This device can provide a drug-free alternative in accelerating healing and decreasing pain post injury [89].

Compared to devices imbedded with emitters, self-emissive devices are intrinsically flexible, advantageous in terms of uniformity and lightweight. For instance, as the next generation of wearable phototherapy devices, OLED can not only provide performance improvements in uniformity and flexibility, but also provide an unprecedented user experience. Recently, a novel wearable patch using a flexible OLED as light source has been manufactured, which can be attached to the body surface as a wound healing platform [87]. The ultralight patch with laminated structure contains the flexible OLEDs, encapsulation layer, heat sink layer and thin film battery. Stacked onto a flexible PET substrate, the OLEDs layer can be fabricated through thermal deposition. Notably, through adjusting the structure and thickness of OLEDs film, the peak wavelength can be freely adjusted within the range of 600–700 nm. OLEDs can be applied in various PBM treatments. As shown in Figure 8d, the flexible OLEDs can be adhered well to human skin as a uniform and size-free surface light source, which can remain thermal stability at temperatures as low as 40 °C during the wound healing process [87]. However, the brightness of existing OLEDs cannot fulfill the condition where light of a specific wavelength has enough tissue penetration. To overcome this disadvantage, quantum dot light-emitting diodes (QLEDs) that enjoy all form factor merits, like OLEDs, have been investigated. In one study, a type of ultrabright and efficient deep red quantum dot light-emitting devices has been demonstrated to have high brightness, which sets a brightness record for existing organic-related red light-emitting devices [86]. For the fabrication process, a polyethylene naphthalate (PEN) film and silicon nitride barrier layer are employed as the substrate. A combination of solution-processing and vacuum evaporation techniques, followed by encapsulation with laminated barrier film, is needed. Ultrabright flexible QLEDs that have great properties, such as high luminance, wavelength tunability and narrow spectra, can be used in photomedicine [86].

## 5. Photomedicine Based on Biocompatible Optical Devices

Biocompatible optical devices combined with sensing modules are undergoing rapid development, which have been applied in clinical diagnosis, therapies and theranostics. All these devices mentioned in the previous sections have diverse photomedical applications. In this section, the photomedical applications of these devices are systematically summarized as optical imaging, physiological signal detection, phototherapy and optogenetics. In the end, the different optical devices and their various applications in photomedicine reviewed from related references are summarized in Figure 9.

### 5.1. Optical Imaging

Optical imaging via biocompatible optical fibers, such as fluorescence and bioluminescence imaging, can realize visualization of tissues and cells with noninvasive, real-time, high-resolution and great temporal-spatial traits [1,90,91]. Optical imaging set-ups, enabling the excitation light delivery, usually integrates the fibers of waveguide for light confinement. Notably, these systems possess the property so that the outgoing light intensity can be achieved through the collection methods based on diverse fibers, such as confocal, dual-clad and external large core fiber, to accurately diagnose the diseases [92,93]. For better analysis of the photo-generated diagnostic information from in-vivo tissues, biocompatible optical fibers with high efficiency of optical propagation have been widely used in fluorescence imaging. Diverse fluorescent proteins, dyes and probes can be doped in the waveguides to increase the quality of fluorescence imaging. Based on the combination of fluorescent signals analysis and simultaneous light therapy, optical devices applied in theranostics can be found [72,94]. Besides, polymeric optical fibers with good resolution of the image reconstruction have been demonstrated, which can transmit the pattern projection and provide the effective assistance in optical imaging [95,96].

### 5.2. Physiological Signal Detection

Human physiological signals contain diverse information that can reveal the health problems; the detection of this information plays an important role in the diagnosis or prediction of diseases. Considering the properties, such as biocompatibility and flexibility to match human tissues, of optical devices, a large number of implantable and wearable biosensors has been investigated in analysis of biomarkers concentration related to various diseases noninvasively or minimally invasively [7,8,16,68,97]. The collection and transmission of the physical and biochemical data can be established on the polymer optical fiber or the compact structure, where biosensors are embedded in electronic tattoos, textiles, wristbands and contact lenses to adapt to the biological environment [39,78,98,99]. The integration of intelligent diagnosis, individualized diagnosis and theranostics with the mobile network is the trend of future development.

#### 5.2.1. Blood Biomarker Detection

Blood biomarkers consist of abundant substances that can sustain the physical and chemical equilibrium of cells, which are significant in disorders assessment. Non-invasive applications of optical devices in blood oxygen sensing, glucose monitoring and heart rate detection are widely investigated. The present pulse oximeter which measures the pulse oximetry level reflecting the acute and chronic respiratory disease is of wearable size [81]. Besides, optical measurements of hemodynamics can avoid physical contact with the skin. The detection is established based on the changes of the contents of hemoglobin in its oxygenated and deoxygenated states. The light intensity detected by PD would change according to the light transmittance and reflection resulting from changed contents of hemoglobin. Accordingly, these optical sensors can product both the oxygen saturation level and blood pulse [70]. During the detection process, highly efficient PLEDs and organic photodetectors (OPDs) can be, respectively, used as light excitation and collection units [96]. Moreover, through Raman and near-infrared spectroscopy, wearable optical devices can monitor blood glucose regardless of poor signal-to-noise ratios [100,101]. Highly flexible polymer optical fibers that can sense stress changes, inducing the optical intensity changes, have been investigated in heart rate detection. Non-invasive blood diagnosis in connection with optical device measurement can provide a great progress in the disease assessment.

#### 5.2.2. Noninvasive In-Situ Body Fluid Analysis

Noninvasive sensing methods which can obviate painful and risky blood sampling have been widely demonstrated in the chemical analysis of biofluids. Since the biofluids contain a mass of compounds, such as protein, peptides, lipids, electrolytes (calcium, magnesium, phosphate, potassium or sodium) and metabolites (glucose, alcohol, lactate or cortisol), real-time and continuous monitoring of the analyte, such as sweat [78,102], tear [77,98], interstitial fluid (ISF) [103] and saliva [104], can enable the understanding of human health and performance. Optical methods including, but not limited to, colorimetry, fluorescence and luminescence are integrated with other microfluidic or electrical modules to form multifunctional systems [7,97].

#### 5.2.3. Health Monitoring

The biomaterial-based optical waveguides can provide a safe and reliable alternative to next-generation, smart healthcare devices [105]. For sensing strain and pressure of the human body, a kind of PDMS-based microfiber worn on the wrist to detect the artery pulse has been devised [106]. This kind of optical sensor can accurately detect the heartbeat and arterial blood pressure. In addition, a textile-based respiratory sensing system is integrated with flexible polymeric optical fibers to measure the respiratory rate [107]. This opens up new possibilities, such as by integrating two or more sensors (e.g., a T-shirt would allow obtaining information not only about the breathing rate, but also about the way of breathing) [107]. Another wearable speckle-based heart rate sensor is presented, maintaining the cost-effective and unobtrusive characteristics described in the literature. A single-board computer (SBC) was chosen along with a CMOS camera sensor for heart-rate monitoring, using POF embedded into a reflective vest. Coherent light from a laser diode directly coupled to the fiber provides the speckle pattern, which is retrieved directly from the fiber end [70].

### 5.3. Phototherapy

#### 5.3.1. PTT

As an adjunct to chemotherapy and radiotherapy for treating cancer, PTT using external photo energy to heat hypoxic tumors with no requirement of oxygen has been developed in recent years. Biocompatible optical devices combined with sensing modules can play an important role in PTT, both as subject and auxiliary tools [108]. Typically, strongly absorbing gold nanocages can induce hyperthermia ablation assisted by a flexible optical fiber needle array. The array incorporates the therapeutic function and sensing ability, which can deliver NIR light into deep subcutaneous tissues for PTT and simultaneously provide real-time imaging feedback [109]. Moreover, optical devices with dopant incorporated have been recently demonstrated to reinforce therapeutic effects, such as a new class of graphene-dissolved PAA microneedles that can be inserted into a specific layer of skin painlessly and generate efficient heat [62]. In summary, highly efficient optical devices transmitting photo energy and a new class of composites of biomaterials and nanoparticles shall be focused on in the future research.

#### 5.3.2. PDT

PDT consisting of three components (a photosensitizer, light energy and oxygen) can be leveraged in combinatorial therapies of PDT and chemotherapy for overcoming resistance to cancer drugs. Through light absorption of specific photosensitizer accumulating around the target diseased cells, highly cytotoxic singlet oxygen and reactive oxide species (ROS) restricting oxygen and nutrient supply can be generated in order to kill tumor cells [110]. Clinically, bladder, brain, esophageal, lung, ovarian and skin cancers can be treated with PDT [2,111]. In order to overcome the drawback of low light penetration depth (<1 cm) through human tissues, a great number of optical methods are clinically used. Importantly, biocompatible optical fibers and waveguides with highly effective optical structures, exhibiting high transparence and low optical loss, can deliver light into deep target regions. Moreover, nanoparticles such as upconversion materials and bioluminescent molecules can be activated by NIR light [112]. Notably, a wireless photonic device used for monitoring the light dose is reported to achieve therapeutic light delivery for cancer PDT [113].

#### 5.3.3. PBM

PBM, also called low-level laser therapy, is usually attributed to that the photons from the red and NIR light transmitting into mitochondria induce enhanced enzyme activity, electron transport, mitochondrial respiration and ATP production. With many clinical applications, including, but not limited to, wound healing, treating inflammation and cutaneous disease, down-regulating osteoclastogenesis and treating various neurological diseases can be seen [114]. As the propagation medium of the photon, optical devices can deliver specific light doses into target regions to activate signaling pathways with little optical loss and great accuracy. In the future, the wearable prototypes interpreting concentration of analytes will be easily integrated with other wearable tools for monitoring physiological parameters such as tissue oxygen saturation (StO_2)_, heart rate, skin temperature and hydration [115]. Recently, a wearable PBM patch that can be attached to the human body for effective PBM anytime, anywhere was developed [87]. From the patch, a possible trend is that the wearable OLEDs system will be used in various therapeutic applications in the future [87]. However, to integrate the diagnostic and therapeutic functionality, much more functional optical device integrated advanced materials and microfabrication technology shall be investigated for the wearable optical devices for PBM.

### 5.4. Optogenetics

Optogenetics is a revolutionary and innovative technique combined with genetic methods and optical techniques in order to stimulate/inhibit cells, control the neural circuits and regulate the neuron activities [12]. Photosensitive proteins, such as channelrhodopsin-2, Archaerhodopsin and halorhodopsin, which are sensitive to specific wavelengths of light, can be expressed on the neuronal membrane by transgenic technology [94]. The using of light stimulation to switch on or off the selective control of neuronal activity quickly ensures the safe possession of high spatial resolution and high specificity in time, thus precise control of target cells can be achieved [116]. In past decades, a series of preclinical experiments on mammals have been successfully demonstrated to realize specific influence on multiple types of neurons cells in different brain regions.

Optogenetics therapy for conditions such as depression [117], chronic pain [118,119] and laryngeal paralysis [120] has been tested. Besides, numerous studies utilizing optogenetics in cardio cerebral diseases have shown great potential for treating cardiac diseases [121,122], epilepsy [123,124,125], Parkinson’s [126] and Alzheimer’s diseases [127]. However, due to the sheer complexity of the human brain, there exists a remote distance of further study for human use. Notably, a viral vector-based optogenetics therapy for treating retinitis pigmentosa using an optogenetics device is processed into clinical application [128,129].

Regarding the of understanding specific neuron types in local circuit functions, simultaneous optical simulation and electrophysiology recording are needed as a class of optrodes [130,131]. Moreover, in order to avoid damage due to a second insertion after that the opsin is implanted into cells successfully, optrodes integrated with microfluidic function have been developed. Typically, designs of optoelectrical stimulators can be divided into waveguide-based optogenetics stimulators [132,133,134,135] and integrated optoelectrical μLED-based probes [136,137,138,139]. Unlike single waveguides coupled with outer light sources, integrated optoelectrical probes usually contain a rigid silicon or flexible substrates incorporated with optical waveguides, recording electrodes and miniaturized LEDs [116]. Monitoring and modulating the diversity of signals used by neurons and glia in a closed-loop fashion is necessary to establish causative links between biochemical processes within the nervous system and observed behaviors. Simultaneous electrical recording and light delivery of the optogenetics probes endow them with the promising potential of theranostic function.

Importantly, the core factors in optogenetics include the optical properties of the devices, the intrinsic biomaterials and the fabrication process of devices. Usual stiff inorganic materials such as silicon [140,141], silica [142] and metals [143] can be structured into fiber, needle, film or probe to be inserted into tissues. However, the high rigidity of these devices may cause damage to surrounding tissues. The flexible polymer-based integrated optogenetics stimulators [50,121] with better biocompatibility and biodegradability will be manufactured to perform optogenetics experiments. Moreover, with the application of gold nanomaterials, QDs and upconversion nanoparticles (UCNPs), noninvasive optogenetics methods can be realized [144]. This technique can promote the clinical research about neurological diseases therapy and provide better understanding of the brain structure and functions.

### 5.5. Summary of the Different Optical Devices and Their Various Applications in Photomedicine

The optical waveguides and integrated optical devices combined with the optical sensing modules that have been reviewed in the previous sections are summarized as in Figure 9, together with their main biocompatible materials and applications in diagnosis, therapies, health monitoring and optogenetics.

From this figure, the following observations can be further made:

Firstly, the optical waveguides with great biocompatibility, biodegradation and mechanical flexibility can get wider applications in photomedicine. Future research directions related to these waveguides may be focused on three aspects (i.e., the development of novel materials, the structure optimization and upgrading of the waveguides, and the fabrication and encapsulation technology of the waveguides). First, the development trend of materials lies in the promotion of light transmission efficiency due to polymer-based waveguides possessing high light transmission loss. Another trend is the research of functional materials for sensing and particle-doped waveguides with novel functions. Meanwhile, for realizing multiple functions such as light delivery and reception, microfluidic channel and potential induction, structure optimization and upgrading of the waveguides shall be focused on in the future. Furthermore, the advancement of manufacturing technology can improve the implementation performance of these waveguides. Besides, the encapsulation technology can enable the integration of versatile modules.

Secondly, compared with the optical waveguides, the integrated implantable optical devices have the advantages of enabling the combination of sensing and therapies. Usually, these devices can be most widely utilized in phototherapy and optogenetics due to the integration of drug-delivery and sensing components. Besides, physiological signal detection and optical imaging can be realized with capabilities of measuring optical parameters. As a building block, complicated implantable photonic or optoelectronic devices will be developed to allow diverse medical applications in future world.

Thirdly, the integrated wearable optical devices which can be worn over the skin have been mainly used in physiological signal detection and phototherapy. Few applications can be seen in optical imaging and optogenetics. In the fields of biomedical imaging such as photoacoustic imaging and optical coherence tomography, wearable optical devices may simplify the imaging system and reduce the system cost. It can be predicted that the research of wearable optogenetics probes will be popular due to the reduction of surgical injury. In summary, miniaturization and flexibility will be strongly preferred for wearable applications of these devices. The future importance will lie in the exploitation of biocompatible materials and the integration of multifunctional components.

## 6. Conclusions

In this work, the new advances in biocompatible optical devices have been reviewed. These optical devices can be categorized into waveguides, integrated implantable and wearable devices. Firstly, the optical waveguides based on inorganic materials and synthetic or natural biomaterials with great optical properties and tissue-adaptable mechanical properties have been discussed. The fabrication methods of these waveguides and applications in optogenetics and biomedical diagnosis, therapies, monitoring and theranostics are also reviewed. Secondly, the integrated light active and light passive devices with multifunctional modules are discussed, and the fabrication process and the operational modes in photomedicine are also introduced. Thirdly, light sensing methods used in the wearable optical devices (i.e., colorimetry, fluorescence and luminescence mechanisms) are introduced. Wearable optical devices for light therapies are also discussed. Details about the fabrication processes and implementations of these optical modules are also introduced. Lastly, the applications of these devices in photomedicine including optical imaging, physiological signal detection, health monitoring, PTT, PDT and PBM, along with the newly emerging optogenetics, are reviewed.

Aiming at achieving clinical diagnosis, treatment and monitoring, the integrity, accuracy and effectiveness of the optical devices must be improved. As a pillar area, synthesis of new biomaterials and improvement of material performance shall be placed at the forefront. Although the implantable and wearable optical devices have been previously developed for the sensation of various physiological signals, multiplexed sensing with a single waveguide device remains challenging. For human use, clinical devices need to integrate much more multifunctional modules to be more biocompatible and intelligent.

## Figures and Tables

**Figure 1 sensors-20-03981-f001:**
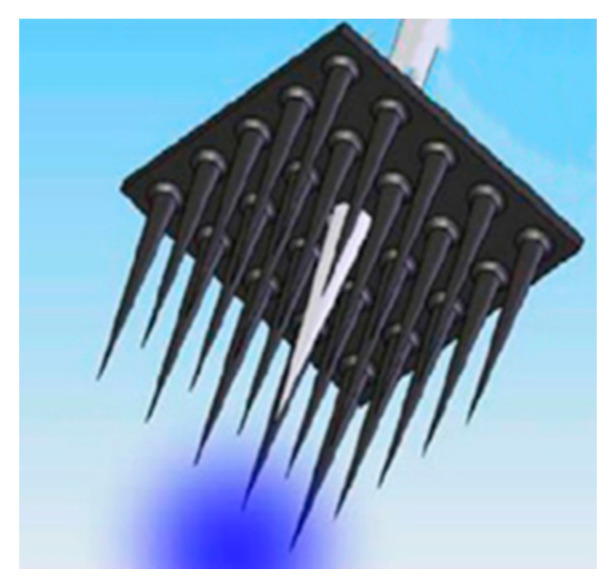
Example of a 3D microfabricated array made of glass.

**Figure 2 sensors-20-03981-f002:**
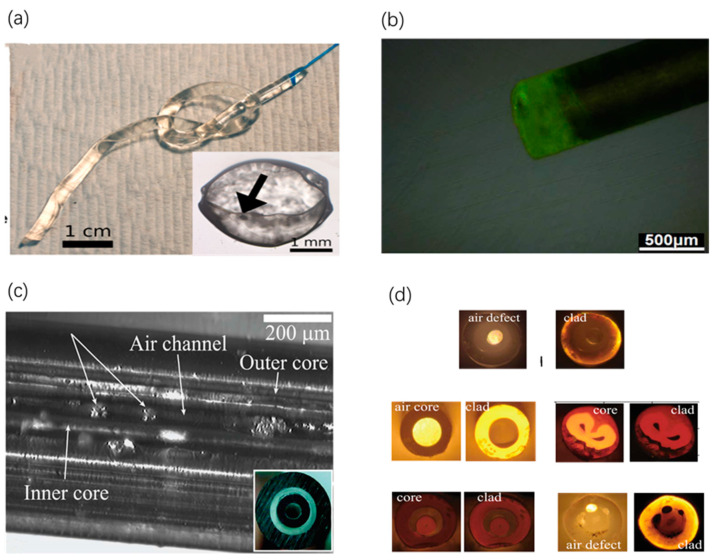
Natural polymer-based waveguides. (**a**) Highly flexible silk waveguide coupled with a glass fiber. Reprinted with permission from [25]. Copyright 2015, OSA. (**b**) Recombinant spider silk optical waveguide with obvious light spot at the terminal. Reprinted with permission from [27]. Copyright 2017, ACS. (**c**) Double-core biodegradable microstructured fiber with a structure of an inner core, air channel and outer core. Inset, preform cross section. The fiber preform [inset] was prepared by using commercially available cellulose butyrate (CB) tubes (refractive index 1.475) of two different diameters. Reprinted with permission from [29]. Copyright 2006, OSA. (**d**) Images of the transmitted light in diverse cross-sections of cellulose-based fibers. Reprinted with permission from [30]. Copyright 2008, SPIE.

**Figure 3 sensors-20-03981-f003:**
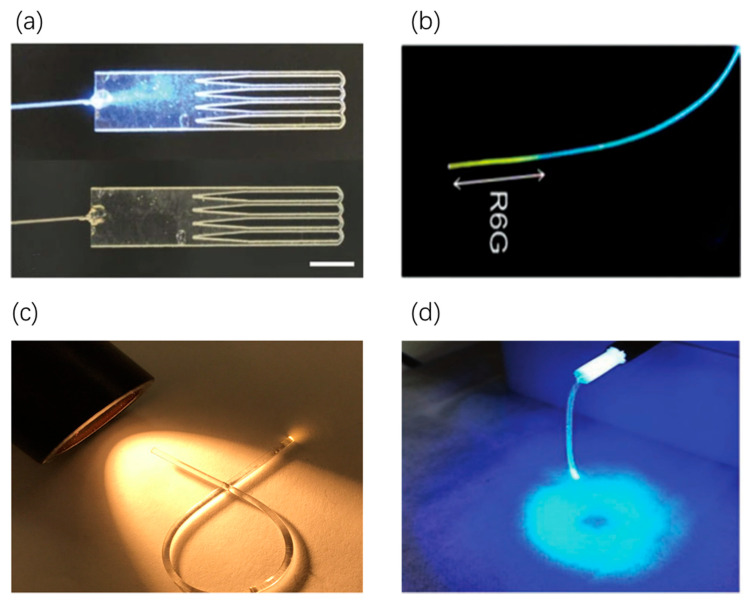
Synthetic polymer-based optical waveguides. (**a**) Comb-shaped slab waveguide. Up—light on; down—off state. Reprinted with permission from [34]. Copyright 2016, Springer Nature. (**b**) A fluorescence image of the fiber doped with Rhodamine-6G. Reprinted with permission from [35]. Copyright 2015, WILEY-VCH. (**c**) A polydimethylsiloxane (PDMS)-based waveguide with a diameter of 1.1 mm. Reprinted with permission from [36]. Copyright 2017, SPIN. (**d**) PAM-based optical fiber coupled with an optical ceramic ferrule and the achieved area of uniform blue light irradiation. Reprinted with permission from [37]. Copyright 2018, WILEY-VCH.

**Figure 4 sensors-20-03981-f004:**
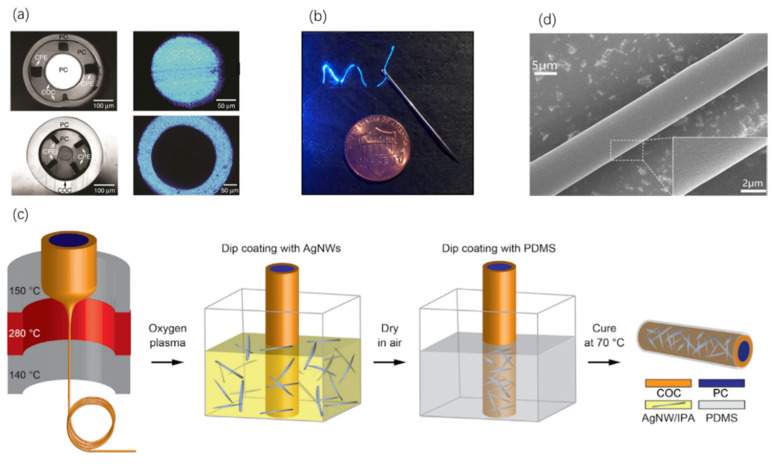
Heterogeneous materials-based waveguides. (**a**) Left: Cross-sections of two designs of PC/COC fiber. Right: Image of light propagation interface profile. Reprinted with permission from [51]. Copyright 2014, Springer Nature. (**b**) Image of probe connected to a laser source, threaded through a needle. Reprinted with permission from [54]. Copyright 2017, AAAS. (**c**) Thermal drawing process of the fiber. Reprinted with permission from [54]. Copyright 2017, AAAS. (**d**) SEM image of QDs-doped polymer microfiber. Inset: Close-up view of the edge. Reprinted with permission from [55]. Copyright 2018, OSA.

**Figure 5 sensors-20-03981-f005:**
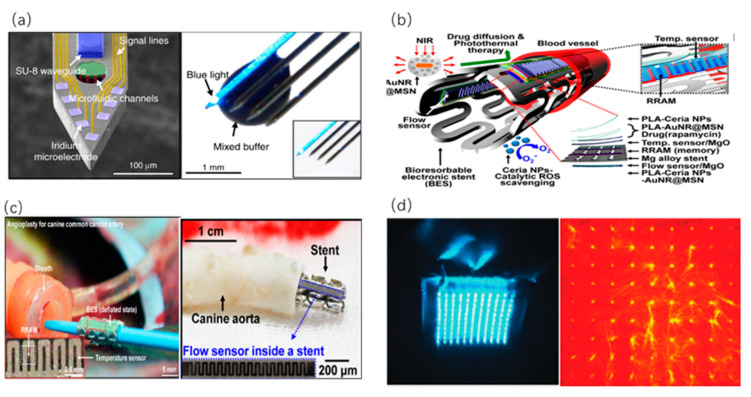
Integrated implantable passivity-based optical devices. (**a**) Left: Composites of top of shank 1. Right: Blue light transmission in SU-8 waveguide. Reprinted with permission from [60]. Copyright 2019, Springer Nature. (**b**) Structures of bioresorbable multifunctional electronic stent integrated with nanoparticles for therapeutic applications. Reprinted with permission from [61]. Copyright 2015, ACS. (**c**) Left: Photographs of the temperature sensor along with the bioresorbable electrical stent. Right: Photographs of the implant in the canine aorta for the ex-vivo experiment of the blood flow sensing. Reprinted with permission from [61]. Copyright 2015, ACS. (**d**) Left: Optical images of collimated 491 nm laser light transmitted through a microneedle array with a micro-lens array. Right: A transmission pattern of blue laser light (491 nm) through an optimally-aligned optical microneedle array (OMNA). Reprinted with permission from [63]. Copyright 2016, OSA.

**Figure 6 sensors-20-03981-f006:**
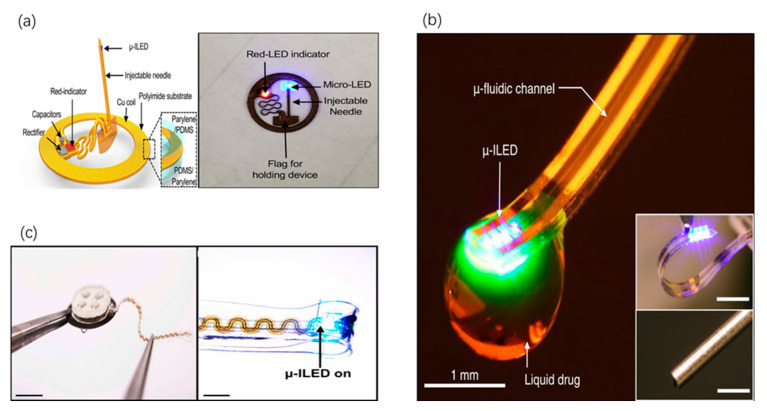
Integrated implantable active optical devices. (**a**) Left: Schematic illustration of the overall construction, highlighting a freely adjustable needle with a m-ILED at the tip end, connected to a receiver coil with matching capacitors, a rectifier, and a separate m-ILED indicator. (Inset) Magnified view of the channels. Scale bar, 100 mm. Right: Representative image of implantable device. Reprinted with permission from [64]. Copyright 2016, Elsevier Inc. (**b**) Optofluidic neural probe during simultaneous drug delivery and photostimulation. (Insets) Comparison of such a device (top) and a conventional metal cannula (bottom; outer and inner diameters of 500 and 260 mm, respectively). Scale bars, 1 mm. Reprinted with permission from [65]. Copyright 2015, Elsevier Inc. (**c**) Left: Demonstrations of the overall size of the system. Scale bars, 5 mm. Right: Magnified views of the neural cuff interface with optical μ-ILED activation. Reprinted with permission from [66]. Copyright 2019, American Association for the Advancement of Science (AAAS).

**Figure 7 sensors-20-03981-f007:**
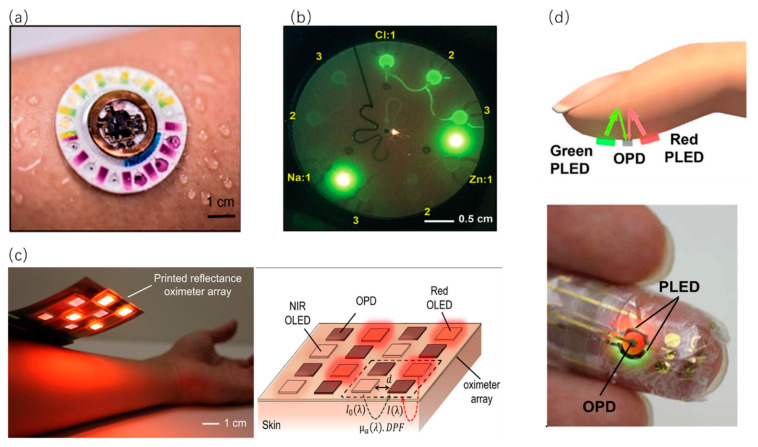
Integrated wearable optical device. (**a**) Image of the device during sweating. Reprinted with permission from [76]. Copyright 2019, AAAS. (**b**) Image of the fluorometric microfluidic device emitted by blue light. Reprinted with permission from [78]. Copyright 2018, RSC. (**c**) Left: Image of reflectance oximeter array (ROA) when in operation. Right: ROA sensor configuration. Red and NIR organic light-emitting diodes (OLED) arrays composed of 2 × 2 pixels each are placed side by side, where the pixels are arranged in a checkerboard pattern. The OPD array composed of 8 pixels is placed on top of the OLED arrays. Reprinted with permission from [80]. Copyright 2018, PNAS. (**d**) Photograph of a finger with the ultra-flexible organic optical sensor attached. Reprinted with permission from [81]. Copyright 2016, AAAS.

**Figure 8 sensors-20-03981-f008:**
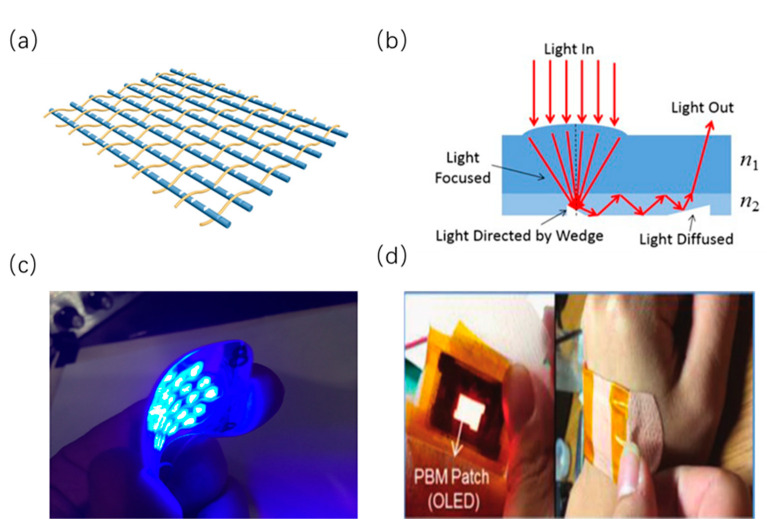
Integrated wearable optical devices for therapy. (**a**) Woven fabrics produced from cotton yarns and polymer optical fibers (POF). Reprinted with permission from [82]. Copyright 2013, OSA. (**b**) An illustration of the light rays propagating through the optical device. Reprinted with permission from [83]. Copyright 2016, SPIE. (**c**) Blue 450 nm printed LED substrate. Reprinted with permission from [89]. Copyright 2019, SPIE. (**d**) Photograph of the patch attached to a human skin. Reprinted with permission from [87]. Copyright 2018, WILEY-VCH.

**Figure 9 sensors-20-03981-f009:**
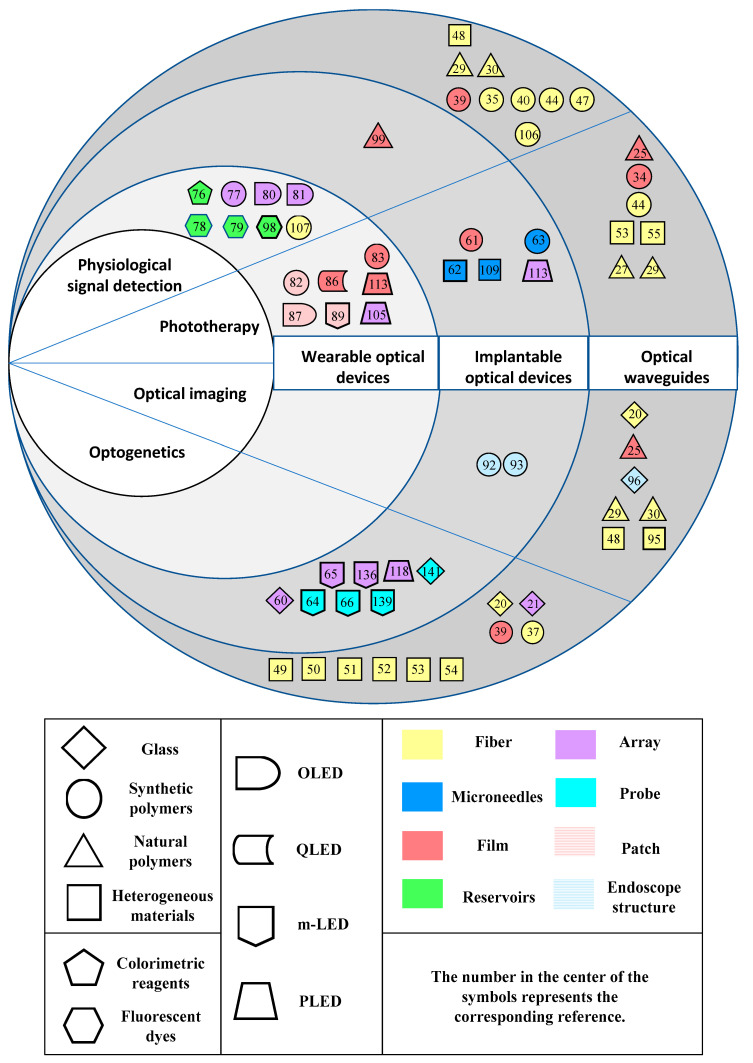
Photomedicine of biocompatible optical devices.

**Table 1 sensors-20-03981-t001:** The optical loss and RI of virous optical waveguides.

Materials ^I^	Interface Profile ^II^	RI ^III^	Optical Loss ^IV^	Ref.
Calcium-phosphate glasses	Core/cladding	1.520/1.527 (633)	0.047 (633)	[20]
Silk fibroin	Core/cladding	1.54/1.34 (532)	1.3–2.7 (540)	[25]
	Core		4.8–6.8 (632.8)	[26]
Recombinant spider silk	Core	1.7 (350–1700)	0.7–0.9 (635)	[27]
Cellulose	Core/double cladding	1.475/1.337 (630)	1–2 (630)	[29]
Cellulose Acetate/PLLA	Core/cladding	1.48/1.45	9.8 (633)	[30]
Cellulose Butyrate	Cladding	1.48	2.2 (633)	[30]
Cellulose Butyrate, PCL	Cladding	1.48, 1.52	6.7 (633)	[30]
Cellulose Butyrate, PCL	Multiple-core/cladding	1.48, 1.52	8.33 (633)	[30]
Cellulose Butyrate, HPC	Porous core/cladding	1.48, 1.34	3.1 (633)	[30]
PDLLA	Core		0.11 (772)	[33]
PLA	Slab	1.47	1.6 (650)	[34]
PEG	Core/cladding	1.351–1.5/1.335–1.339 (532)	1–6 (532)	[39]
PEGDA	Core	1.35	<1	[35]
PDMS	Core/cladding	1.55/1.41	0.36 (635)	[43]
PDMS	Taper core	1.42		[45]
PAM/Alginate	Core/cladding	1.344–1.356	0.2–0.8 (472)	[46]
PC/COC	Core/cladding	1.58/1.52	2.44 (473)	[50]
PC/COC,CPE(design I)	Core/cladding		2.7 (473)	[51]
PC/COC,CPE(design II)	Core/cladding		1.6 (473)	[51]
SEBS/Geniomer	Core/cladding	1.52/1.42	0.74 (465)	[53]
PC/COC, PDMS, AgNW	Core/cladding	1.58/1.52	1.9 (473)	[54]
PS/PMMA	Core/cladding	1.59/1.49 (325)		[55]

^I^: “A/B”—“A” stands for the core material, “B” means the cladding materials. “C + D” means that “C” is the main material; and “D” is the dopant powder. ^II^: “Core/cladding” represents the optical interface of fibers. ^III^: “A/B©”—“A” represents the RI of core; B means the RI of cladding; “C” in the bracket represents the measuring wavelength in nm for optical fibers. “A + B”—“A” is the RI of core; and “B” is the RI of dopant powder. ^IV^: “A (B)’—“A” means the optical loss of waveguide; and “B” is the measuring wavelength in nm.

**Table 2 sensors-20-03981-t002:** Mechanical properties, fabrication process and applications of optical waveguide.

Materials	Mechanical Properties	Key Fabrication Process	Applications	Ref.
Calcium-phosphate glasses	Hard	Perform drawing, rotational casting	PDT, optogenetics and biosensing	[20]
Silica	Hard	Micromachining	Optogenetics, PDT and imaging	[21]
Silk fibroin	Flexible, elastic	Mold casting, drying, gelation	Optical imaging and therapy.	[25]
Silk fibroin	Soft	Femtosecond direct laser writing	Light delivery	[26]
Recombinant spider silk	Good bending resistance	Genetic engineering and mold casting	Light delivery	[27]
Recombinant SELP	Stiff	Genetic engineering and spin	Drug delivery and wound healing	[28]
Cellulose acetate/PLLA	Flexible	Dissolving, co-rolling and thermal drawing	light delivery	[30]
Cellulose butyrate and PCL	Flexible	Powder filling and thermal drawing	Light delivery or a controlled drug delivery	[30]
Cellulose butyrate	Flexible	Power filling, thermal drawing and casting	In-vivo sensing and drug delivery	[30]
PDLLA	Stiff	Mold melting, heat drawing	PDT	[33]
PLA	Stiff	Melt pressing, solvent casting and ultraviolet-induced crosslinking techniques	Health monitoring, controlled drug release and chronic PDT	[34]
PEGDA	Flexible	UV induced polymerization and crosslinking	Optogenetics and cell encapsulation	[39]
PEG	Flexible	Photo crosslinking and dip-coating	Fluorescence and photomedicine	[35]
p(AM-*co*-PEGDA) and Ca alginate	Flexible	UV-induced mold polymerization and dip-coating	Glucose sensing	[40]
PEGDA	Soft	Photopolymerization	Health monitoring	[41]
PDMS	Stretchable, flexible	Mold curing and dip-coating	Optical sensing	[43]
PDMS	Highly stretchable and soft	Curing, coating and covering	Pressure, strain, and curvature sensing	[44]
PDMS	Flexible	Mold curing	Light delivery	[45]
PAM		Photo cross-linking, deposition and silanizing	Wound healing monitoring	[46]
PAM/Alginate	Flexible, high-stretchable	Photo cross-linking and silanizing	Optogenetics	[37]
PAM Au nanorods		Direct drawing and deposition	Relative humidity (RH) sensing	[47]
PC/PMMA, PSU and CPC	Flexible	Rolling and thermal drawing	Light health care and fluorescent imaging	[48]
PC/COC	Stiff	Mold casting and thermal drawing	Optogenetics	[50]
PC, COC and CPE	Soft	Thermal drawing	Optogenetics and drug delivery	[51]
PEI, PPSU and Sn	Soft	Thermal drawing	Optogenetics and drug delivery	[51]
SEBS/Geniomer	Highly stretchable	Thermal drawing	Optogenetics and light therapy	[53]
PC/SEBS, PDMS and AgNWs	Flexible	Thermal drawing and dip-coating	Optogenetics and other health-care	[54]
PS/PMMA	Flexible	Thermal drawing	Photomedicine	[55]

**Table 3 sensors-20-03981-t003:** Optical modules and fabrication process of integrated implantable optical devices.

Light Source	Optical Sensing Modules	Key Fabrication Process	Applications	Ref.
Coupled optical fiber	SU8/glass waveguide	Photolithography, wet etching	Optogenetics	[60]
Outer LED/therapeutic nanoparticles	Stent	Photolithography and Reactive Ion Etching	PTT, physiological signal detection	[61]
Outer laser diode	Microneedles	Casting	PTT, drug delivery	[62]
A micro-lens array coupled with LED	Microneedle arrays	Press melting	Drug delivery, PDT	[63]
m-ILED	Needle	Electro-deposited, photolithographic technique, etching, laser cutting and dip-coating	Neural stimulation	[64]
m-ILED array	Waveguide	Mold curing and pressing	Optogenetics	[65]
μ-ILED	Cuff	Thermal drawing, corona treatment and pre-strain releasing	Optogenetics	[66]

**Table 4 sensors-20-03981-t004:** Integrated wearable optical devices for diagnosis.

Optical Methods	Optical Sensing Modules	Key Fabrication Process	Applications	Ref.
Colorimetry	Reservoirs storing dyes	Casting	Chloride, glucose, lactate and pH measurements	[75]
	Colloidal crystal array connected with lens	Mold casting	Glucose concentration test	[76]
	Array chambers with adsorbent-based sink	Casting	Sweat pH level	[77]
Fluorescence	Sweat fluids and fluorometric array	Photolithography	The cystic fibrosis diagnosis	[78]
	Containment reservoirs inserted with reagents	Soft lithographic	Lactate, pH and glucose detection	[79]
Luminescence	OLED accompanied by two PLEDs array	Press melting	Pulse oxygenation measurement	[80]
	OLEDs and NIR OLEDs arrays	Etching, laser cutting and dip-coating	Oximeter	[81]

**Table 5 sensors-20-03981-t005:** Integrated wearable optical devices for therapy.

Light Sources	Optical Sensing Structures	Key Fabrication Process	Applications	Ref.
Coupled LEDs	V-grooved step-index fibers	Saw, weave	Wound healing	[82]
LED arrays	PDMS-sheet	Soft-lithography techniques	Phototherapy	[83]
Printed LEDs	Polyester film	Roll-to-roll printed micro LED technology	PBM	[89]
OLEDs	PET substrate	Thermal deposition	Wound healing	[87]
QLEDS	PEN film and silicon nitride layer	Solution-processing, vacuum evaporation techniques	Phototherapy	[86]

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
