# Peer review of "Optical Waveguides and Integrated Optical Devices for Medical Diagnosis, Health Monitoring and Light Therapies"

_sensors, 2020, doi:10.3390/s20143981_

Round 1

Reviewer 1 Report

I think a review paper should serve both as an introduction to a field for the non-expert reader, and to draw a complete state of the art picture of that field for the expert reader. Now, as a non expert reader I am not entitled to argue about the completeness of the picture here presented, so I will only report my impression as a reader working in the optical sensing and willing to learn something new about the biomedical applications.

In the first part of the manuscript (where the biocompatibility problems of silica fibers and all the alternative waveguide designs are described) I found many useful and well organized informations. few comments:

Line 214: what is low-level light?

Line 299: what is the connection between localized surface plasmon resonances and respiratory diseases? I know there is a reference, but these kind of sentences thrown there without explanations make the manuscript hard to read.

Line 326 “of perform”?

In the second and third part of the manuscript (implantable and wereable optical devices), I found the informations much more vague and I couldn’t gather the same level of understanding. A huge amount of fabrication technical details are given for the presented devices (In many cases with a large use of acronyms not previously introduced), but their principles of operation are rarely described. Main points to check:

1) “implantable active optical devices” section (from line 475)...there is a huge digression on the fabrication technique of this “complicated” (!) device that delivers a “light simulation” (!). Did not get anything here.

2) in the “wearable optical devices” section. Line 587: “the diffracted color changes”. The color diffracted by what? there must be a grating but it is not described...And the light source, and the detector?

3) lines 639-647: pulse oxygenation measurement device: why does it use two light sources at two different wavelengths (red and green)? 

Other typos :

737-740 please rephrase.

765 “as well as”….don’t understand this sentence.

771 “Combined” also don’t get this sentence.

781 “virous” would be various or virus?

792 “the present pulse oxymeter”, which is?

795 “the basis of change of hemoglobin in its hoxigenated and non oxigenated states”. What changes?

857 “photos” must be a typo

In general, I recommend the authors to revise the manuscript aiming at:

1) More sinthesys. I found the manuscript too verbose and longer than necessary.

2) try to better balance the informations they convey: huge level of details in the fabrication techniques, quite vague on the principles of operation and on the applications of the devices described.

3) check that all the acronyms they use are defined first.

4) check for typos: there are many.

Reviewer 2 Report

This submission has provided a comprehensive review on the topic of integrated optical devices for biomedical applications. Given the rapid expansion of photonics into the healthcare related disciplines in recent years, this submission should indeed present a timely case for readers’ quick assessment of the current situation. To improve the quality of this submission, I have the following suggestions:

1) Add a table of contents to help readers appreciate what topics have been covered.

2) Why current waveguides used in other areas, e.g. telecom, cannot be directly adopted? There are more unique reasons to go for “biocompatible waveguides”.  A additional paragraph at the start of “2 Biocompatible optical waveguides” to address this point is recommended.

3) Figure 9 provides a convenient summary on current status. Figure 9 also highlights possible gaps that are yet to be fully explored. The authors may wish to take one step forward to add a few paragraphs to list out those gaps and identify possible development trends and future directions.
